# Seasonal Variation in Recovery Process of Rainwater Retention Capacity for Green Roofs

Yinchao Hu [1,2], Huapeng Qin [1,*,†], Yiming Ouyang [1] and Shaw-Lei Yu [3]

1 Key Laboratory for Urban Habitat Environmental Science and Technology, School of Environment and Energy, Peking University Shenzhen Graduate School, Shenzhen 518055, China
2 Department of Natural Resources & Environmental Sciences, University of Illinois Urbana-Champaign, Urbana, IL 61801, USA
3 Department of Engineering, Systems and Environmental Engineering, School of Engineering and Applied Science, University of Virginia, Charlottesville, VA 22908, USA
* Correspondence: qinhp@pkusz.edu.cn; Tel./Fax: +86-755-26035291
† Current address: Room 414, E Building, Peking University Shenzhen Graduate School, Lishui Road, Xili, Nanshan District, Shenzhen 518055, China.

**Highlights:**

The recovery process of the retention capacity is described using an exponential decline curve. The recovery rate is represented by the half-life of the water content after a storm ($T_{50}$), and $T_{50}$ varies with the seasonal variations in the weather conditions and vegetation coverage. During the wet season, $T_{50}$ affects the rainwater retention performance of green roofs; during the dry season, the actual retention capacity, rather than $T_{50}$, affects the performance.

**Abstract:** Green roofs need to quickly recover their water retention capacity between rainfall events to maintain their rainwater retention performance. In this study, the authors observed the rainwater retention, recovery process, and plant eco-physiological performance of green roofs with four local vegetation species under a typical subtropical monsoon climate for two years. The half-life of the water content after each rainfall ($T_{50}$) was used to describe the recovery rate. The results indicate that (1) the decline in the water content after rainfall can be well described by an exponential decline curve ($R^2 > 0.7$), and the average $T_{50}$ of green roofs with *Plectranthus prostratus* Gürke was the shortest among the four plants; (2) the T50 in the wet season was significantly shorter than that in the dry season ($p < 0.01$) because of the seasonal variations in the weather conditions and eco-physiological activity, such as vegetation coverage and transpiration; (3) the rainwater retention of green roofs for rainfall events in the wet season was significantly lower than that in the dry season due to a relatively short antecedent dry period; (4) plants with a high maximum photosynthetic capacity, a strong root system, drought resistance, and large vegetation coverage were recommended as green roof plants. Above all, *P. prostrates* was found to be the best choice in the study.

**Keywords:** green roof; rainwater retention; recovery process; seasonal variation; weather condition; plant eco-physiology

## 1. Introduction

With the acceleration of urbanization, the urban impermeable surface has increased massively, which has led to increases in rainfall runoff and the risk of urban flooding [1–4]. Green roofs have the ability to retain rainwater during rainfall, and thus, reduce runoff. They are recognized as part of a resilient stormwater management practice and have been widely used in urban areas, particularly in urban areas with limited space and high density [5–9]. When green roofs are designed to reduce rainfall runoff, it is necessary to consider not only the rainwater retention performance during a single rainfall event but also during a long period of alternating wet and dry conditions. This requires green roofs to quickly

recover their water retention space during dry periods between rainfall events [10–13]. Understanding the recovery of retention performance can help explain the variation of rainwater retention and provide guidance for green roof design and management.

The recovery of the rainwater retention performance of green roofs mainly depends on the evapotranspiration (ET) during dry periods, which is an important part of water budgeting [14,15]. The daily ET rate was used to represent the recovery rate during dry periods [10]. Kemp et al. [16] used the total ET amount during the 72 h after rainfall to characterize the recovery rate of the water retention capacity in green roofs during dry periods. However, as a dynamic process, the ET rate fluctuates greatly and is difficult to quantify [17]. Berretta et al. [5] predicted the ET by using the basic form of the soil moisture extraction function (SMEF) model based on the actual moisture content, calculated PET, and water balance. In addition, an accurate long-term hydrological model of green roofs needs to consider the complete infiltration, runoff, ET in the water storage layer, and the surface ET [18]. Therefore, temporary variation in the moisture contents of green roof substrates can directly reflect the recovery of the rainwater retention capacity.

Previous studies have reported that the recovery of the rainwater retention performance is affected by internal factors such as substrate characteristics and plant species. For example, the ET of green roofs highly depends on the water content in the substrate, and the ET rate is lower under limited moisture conditions [5,10,11,16]. Adding a storage layer at the bottom of green roofs can help supplement water to the substrate layer and thus improve the ET, especially in warm and dry seasons [18,19]. Plant transpiration is also considered to be an important control factor of the ET rate [11,20,21]. The presence of vegetation in green roofs maintains a high daily water loss after several days of drying [5]. In addition, the ET can be affected by plant coverage. The greater the plant coverage, the greater the contribution of plant transpiration to the ET [22,23]. The types of green roof plants also have a significant effect on the ET rates [16,20]. The climate adaptability, photosynthesis rate, water demand, pollution resistance, and planting cost should be considered when selecting the proper plant species [24]. Farrell et al. [25] investigated the water-use strategies of 12 species of different life forms (monocot, herb, and shrub) and indicated that most species achieved moderate to high transpiration rates, using substantially more water than succulent plants.

The recovery rate of the rainwater retention performance of green roofs is also affected by external weather conditions [11,16,26,27]. The impact is due to the dynamic changes in the rainfall and ET influenced by solar radiation, wind speed, air temperature, and relative humidity [28]. For example, Sims et al. [29] reported that green roofs in arid or semi-arid climates had a higher ET and higher average annual rainfall retention than those in temperate and maritime climates. Some harsh environmental conditions, such as drought and high temperature stresses, will decrease the ET rate of green roofs by affecting the plant physiological state [29,30]. Most previous studies have shown that the ET rate is higher in warm and hot summers, reflecting strong seasonal differences [5,11,27,31]. The results above partly explain the geographical and seasonal differences in the ET and retention performance of green roofs. Thus, it is difficult to evaluate the performance of green roofs based on previous studies in other regions [32].

There is an almost general consensus that green roofs with a higher recovery rate can achieve higher rainwater retention [10,11,16,18]. However, previous studies have hardly obtained long-term rainwater retention performance observation data and have usually utilized hydrological models for simulations. During a long-term observation, the weather conditions and eco-physiology of the plants will change in different seasons. Few studies have been carried out on the influence of vegetation types and their eco-physiology variation on the recovery process, particularly under different weather conditions in different seasons. It is also necessary to further investigate the effect of the recovery rate on the rainwater retention of green roofs under different seasonal conditions.

In this study, observations of the recovery of the rainwater retention performance for green roofs with four vegetation species, including *Callisia repens* Linnaeus, *Portulaca*

*grandiflora* Hook., *Plectranthus prostratus* Gürke, and *Sedum lineare* Thunb., were carried out in Shenzhen, China. The recovery process and the recovery rate of the rainwater retention performance of the green roofs were respectively described by an exponential decline curve and the half-life of the water content in the green roofs after a rainfall event ($T_{50}$). The objectives of this study were to (1) evaluate the recovery of the rainwater retention capacity of green roofs; (2) analyze the seasonal variation in the recovery with four vegetation species; and (3) understand the effect of the vegetation species with different recovery rates on the rainwater retention capacity.

## 2. Materials and Method

### 2.1. Experimental Setup

A green roof experiment site was set up on a building top at the campus of the Peking University Shenzhen Graduate School (PKUSZ) (22°35′48.26″ N, 113°58′22.70″ E), China. The study site has a typical subtropical monsoon climate with a mean annual temperature of 22 °C and a mean annual rainfall of 1944 mm. Most of the rainfall in Shenzhen is concentrated during April–September, accounting for 80% to 90% of the annual rainfall. The seasons can therefore be divided into a wet season, with hot and rainy weather (April–September), and a dry season, with mild and dry weather (October–March).

The experiment platform (Figure 1a) included eight acrylic glass material green roof modules (four plant type treatments with two replicates per treatment) with the dimensions of 1 m × 1 m × 0.2 m (length × width × height). Each module had four layers from top to bottom: a vegetation layer, a substrate layer, a geotextile layer, and a storage layer (Figure 1b). In the vegetation layer, four different plants were set as the only variables in the experiment, namely *Callisia repens* Linnaeus (*C. repens*), *Portulaca grandiflora* Hook. (*P. grandiflora*), *Plectranthus prostratus* Gürke (*P. prostrates*), and *Sedum lineare* Thunb. (*S. lineare*). All four plants have a conservation water-using strategy: *C. repens*, *P. prostrates*, and *S. lineare* are crassulacean acid metabolism (CAM) plants, which open their stomata at night and close them during warmer days to retain moisture [20,33,34], while *P. grandiflora* is a type of C4 plant, but it has low stomatal conductance and a relatively low transpiration rate [35]. Therefore, the four plants are used extensively for green roofs in South China. The substrate layer was filled with engineered light soil (consisting of organic peat, perlite, and vermiculite at a 5:3:2 ratio) to a depth of 10 cm. The soil organic matter content was 280 g/kg, and the soil porosity and field capacity were 70% and 57%, respectively. The geotextile layer was set under the substrate layer to prevent the soil from entering the storage layer. The storage layer was 3 cm deep, consisting of a storage drain plate (2 cm) and ceramicites (1 cm). After the platforms were built, the green roofs only received natural rainfall, without additional irrigation or fertilization measures.

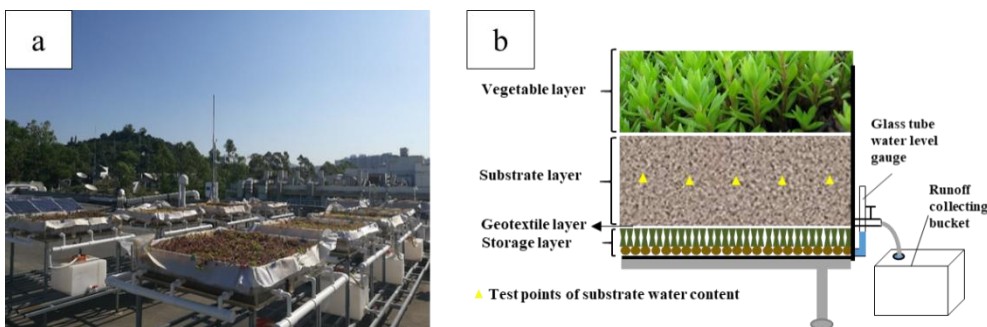

**Figure 1.** The platform for the green roof experiment. (**a**) Overall layout of the green roof platform; and (**b**) structure of the green roof module.

*2.2. Data Collection and Analysis*

2.2.1. Meteorological Data and Rainfall Events

The meteorological data were measured by an automatic weather station (Davis wireless vantage Pro2 Plus, Hayward, CA, USA) installed at PKUSZ. The rainfall amount was recorded at intervals of 10 min. Other parameters, including solar radiation, wind speed, air temperature, and relative humidity, were recorded at intervals of 1 h.

The monitoring experiment was conducted from August 2018 to December 2019. A total of 60 rainfall events (with a rainfall depth > 0.2 mm and ADP > 6 h) were recorded during the monitoring period. In the wet season, there were 46 rainfall events, the maximum rainfall and the average rainfall were 87.5 mm and 25.7 mm, respectively, and the maximum ADP and average ADP were 266.3 h and 52.2 h, respectively. In the dry season, there were 14 rainfall events, the maximum rainfall and the average rainfall were 45 mm and 10.3 mm, respectively, and the maximum ADP and average ADP were 528 h and 114.8 h, respectively.

2.2.2. Daily Water Content

The water content of green roofs consists of the moisture content in the substrate layer and the water content in the storage layer. The moisture content was measured by a portable hand-held soil moisture sensor (SM300, Delta-T, Cambridge, UK) at 5 points, which were 5 cm deep and evenly distributed horizontally in the substrate layer (3 times for each point). The moisture content of the substrate layer was determined by the average of the measurements. The water depth in the storage layer was measured by a water level gauge of a glass tube connected to the bottom of the layer. The water content in the layer was estimated by multiplying the water depth and the porosity of the layer. Both the moisture content in the substrate layer and the water content in the storage layer were recorded at intervals of 24 h.

2.2.3. Recovery Process and Recovery Rate

The decline of the water content during a sunny period reflects the recovery process of the rainwater retention capacity of green roofs. The daily water content after the rainfall events was normalized, and the scatter plot of the water content changing with days can be made. This scatter plot was further fitted by nonlinear regression equations as a recovery curve. According to the comparison of the goodness of fit between the common decline curves (exponential decline curve, hyperbolic decline curve, and harmonic decline curve), the exponential decline curve was selected to fit the measurement data (with the largest value of $R^2 > 0.7$). This suggests that the water content after rainfall declined exponentially with the available recovery days. The equation is as follows:

$$\theta_t / \theta_0 = e^{-k \cdot t} \tag{1}$$

where $\theta_0$ is the initial water content after rain; $\theta_t$ is the water content t hours after rain; and k is the recovery coefficient of the recovery curve.

$$T_{50} = (\ln 2)/k \tag{2}$$

where $T_{50}$ is the half-life of the water content in the green roofs, which can be defined as the time taken for the water content of the green roofs to fall to half of its initial value after a rain (day). In this study, $T_{50}$ was used to measure the recovery rate of the rainwater retention capacity of the green roofs.

2.2.4. Runoff and Rainwater Retention

An outlet was installed at the top of the storage layer of each green roof module. During rainfall events, once the storage layer was full, the runoff was generated and discharged from the outlet. A bucket with a capacity of 65 L was used to collect and measure the runoff volume of each rainfall event. The rainwater retention performance of the green roofs for each rainfall event was estimated by subtracting the runoff volume from

the rainfall amount. The retention rate was calculated as the ratio of the retention volume to the rainfall volume.

Generally, the available water retention capacity of the soil can be calculated in the laboratory using the volumetric water content at the field capacity minus the volumetric water content at the permanent wilting point [36]. However, the actual water retention capacity is usually less than the available capacity [37]. In this study, the rainfall events that met the following requirements were chosen: (1) the initial moisture content of the substrate layer of the green roofs was low enough (<5%, close to the wilting point) (2) the rainfall amount was large enough to generate runoff from the green roofs. The average rainwater retention of the green roofs for these rainfall events was taken as the actual retention capacity.

### 2.2.5. Plant Eco-Physiology

The plant coverage was measured once per week using a digital image analysis (DIA) method during the experiment [33]. To minimize the effect of shadows, the measurements were taken at noon (from 11:00 a.m. to 13:00 p.m.) and from a perpendicular position.

The plant maximum photosynthetic capacity (Fv/Fm) was measured by pulse amplitude modulation (PAM-2500, Walz, Effeltrich, Germany). The slow light reaction curve was selected, and the measurement time was from 8:00 p.m. to 11:00 p.m. The measurement was repeated 3 times and the average value was recorded as Fv/Fm. The indicator was measured for 3 days during both the wet season and the dry season.

### *2.3. Statistical Analysis*

All statistical analyses were carried out using SPSS 26.0 edition software (IBM, New York, NY, USA). Two-way analysis of variance (ANOVA) was used to assess the effects of the vegetation types and seasonal variation on the recovery rate and retention performance. A paired *t*-test was used to analyze the differences in the actual retention capacity among the four vegetation species. The least significant difference (LSD) at a 0.05 significance level was used to detect the differences between treatment means. Correlation analysis was used to determine the strength of the relationship between the recovery rates ($T_{50}$) of the different vegetated green roofs and the air temperature, relative humidity, wind speed, and radiation. It was also used to determine the relationship between the retention performance of the different vegetated green roofs and the antecedent dry period, antecedent moisture content, rainfall depth, rainfall duration, and rainfall intensity. A *t*-test was used to analyze the differences in the actual retention capacity among the four vegetation species.

### 3. Results and Discussion

### *3.1. Recovery of the Retention Capacity of Green Roofs*

In order to analyze the recovery of the rainwater retention capacity, the measured water contents during the dry periods of all single rainfall events in the monitoring period were fitted, and the $T_{50}$ for each dry period was calculated, as shown in Figure 2. By normalizing and fitting all data of the daily water contents after rainfall events during the whole monitoring period with Equation (1), the curves of the recovery process of the green roof retention capacity were obtained (Figure 3). The average, maximum, and minimum $T_{50}$ of the green roofs with different vegetation types had ranges of 4–6 days, 7–17 days, and 2–3 days, respectively. All four plants had a significant fluctuation in the $T_{50}$ during the monitoring period. In addition, there were obvious differences in the $T_{50}$ of some green roofs with different vegetation ($p < 0.05$). The modules planted with *P. prostrates* and *S. lineare* had the fastest recovery rates (with average $T_{50}$ values of 4.3 days and 4.7 days), while the modules planted with *C. repens* and *P. grandiflora* had the slowest recovery rates (with average $T_{50}$ values of 5.5 days and 5.9 days). The order of the average recovery rates of the green roofs with different vegetation types from fastest to slowest was: *P. prostrates* > *S. lineare* > *C. repens* > * *P. grandiflora* (>* means significantly greater). The

*k* coefficient of the recovery curve (Figure 3, the black curve) had the same recovery rate order between treatments.

　　It was necessary to further analyze the factors affecting the recovery process and the reasons for the differences among the different plants in the recovery process of the retention capacity. The correlation analysis shown in Table 1 indicates that the $T_{50}$ values of the green roofs with different vegetation types were significantly negatively correlated with the average solar radiation intensity rates in the dry period ($p < 0.05$) because solar radiation provided the most energy for the ET, and thus, for the recovery of the rainwater retention capacity. In addition, the $T_{50}$ values of the green roofs with *P. grandiflora* were significantly negatively correlated with the air temperature ($p < 0.05$). In addition, there was no significant correlation between the recovery rate and other meteorological parameters, e.g., relative humidity and wind speed. Therefore, the recovery rates of the green roofs with different vegetation types were mainly affected by the intensity of solar radiation.

　　The differences between the different plants can be explained by plant eco-physiology characteristics. As shown in Figure 4a, the order of the maximum photosynthetic capacity was: *P. prostrates > S. lineare > P. grandiflora > C. repens*, which is exactly consistent with the order of the recovery rate during the whole monitoring period. This is because the photosynthetic capacity reflects the stomatal movement of plant leaves, which is the determining factor of the transpiration rates of plants [38,39].

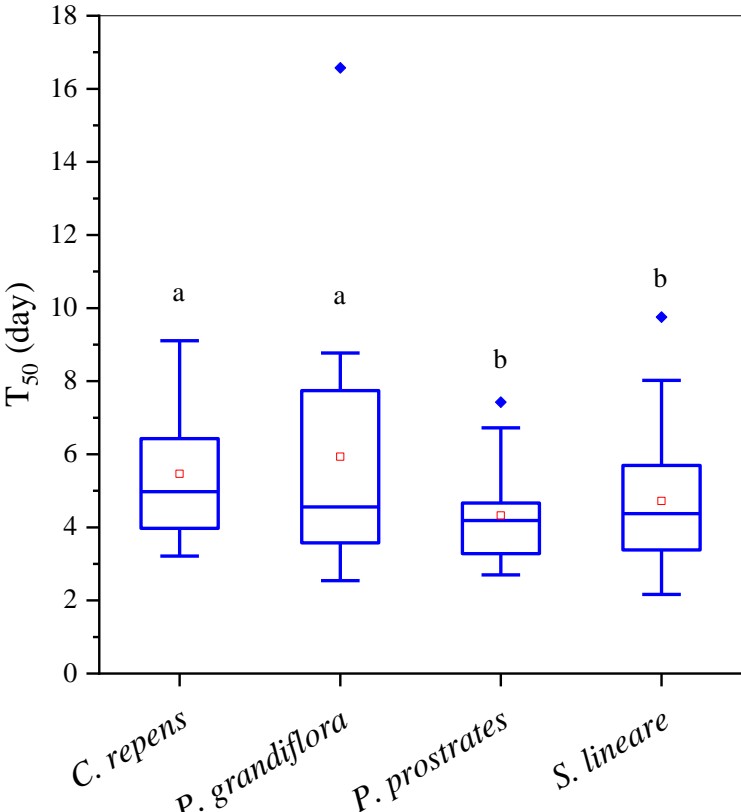

**Figure 2.** Recovery rate ($T_{50}$) for all plant treatments during the whole monitoring period. The centre horizontal line in each box of the boxplots is the median value, and the red square is the mean value. The letters at the tops of the boxes showed the significances at the 0.05 level (two-way ANOVA without interactions).

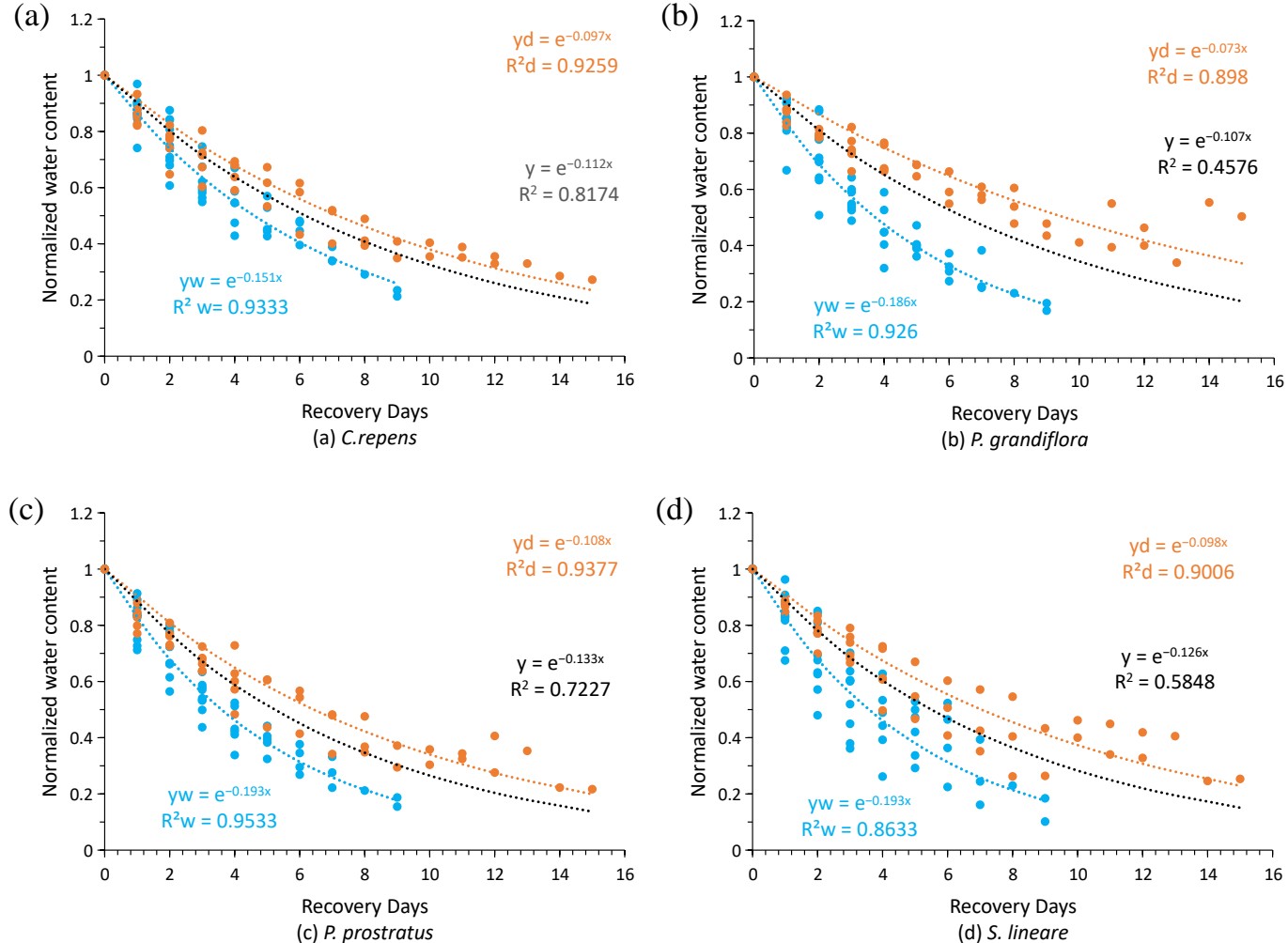

**Figure 3.** Recovery curves of green roofs for all plant treatments fitted by normalized water content after rainfall: (**a**) *C. repens*, (**b**) *P. grandiflora*, (**c**) *P. prostrates*, and (**d**) *S. lineare*. (Orange curve: dry season; blue curve: wet season; black curve: all data).

**Table 1.** Correlation coefficients between $T_{50}$ of different vegetated green roofs and air temperature, relative humidity, wind speed, and radiation.

| Plant Type | Temperature | Relative Humidity | Radiation | Wind Speed |
|:---:|:---:|:---:|:---:|:---:|
| *C. Repens* | −0.38 | −0.67 | −0.75 * | −0.33 |
| *P. Grandiflora* | −0.81 * | −0.39 | −0.90 ** | −0.40 |
| *P. Prostrates* | −0.58 | −0.48 | −0.91 ** | −0.41 |
| *S. Lineare* | −0.38 | −0.60 | −0.77 * | −0.59 |

Note: * $p < 0.05$. ** $p < 0.01$.

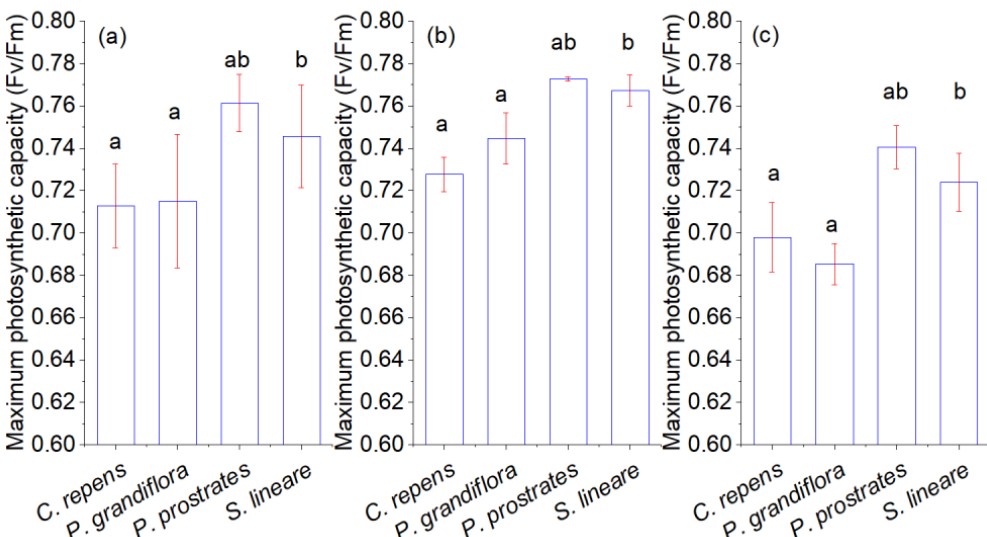

**Figure 4.** Maximum photosynthetic capacity (Fv/Fm) for all plant treatments during (**a**) the entire monitoring period, (**b**) wet season, and (**c**) dry season. Bars are the standard deviation of the mean. The letters at the tops of the columns showed the significances at the 0.05 level (paired *t*-test).

*3.2. Seasonal Variation in the Recovery of the Retention Capacity of Green Roofs*

According to Section 3.1, the recovery process was affected by solar radiation and the potential photosynthetic capacity of plants, so the changes in the meteorological conditions and plant physiological activity in different seasons were bound to affect the recovery process of the green roofs. As shown in Figure 3, two recovery curves of the green roofs in the wet season (blue) and dry season (orange) in Shenzhen were obtained by the normalization and fitting of the measured water content changes after rainfall. After distinguishing the wet season from the dry season, the fitting degree of the fitting curve was greatly improved ($0.8 < R^2 < 0.95$). The curves showed significant seasonal differences in the recovery process and obviously higher recovery rates in the wet season for all species treatments.

As shown in Figure 5, the $T_{50}$ values in the wet season were significantly shorter than those in the dry season ($p < 0.01$). As for different species treatments (Figure 5a,b), there were also significant differences in the recovery rate and recovery rate variation between seasons ($p < 0.05$). The fitting $T_{50}$ of the modules planted with *C. repens*, *P. grandiflora*, *P. prostrates*, and *S. lineare* were 4.6, 3.8, 3.5, 3.6 days, respectively, in the wet season, and 7.0, 9.1, 5.5, and 6.3 days in the dry season. The order of the average recovery rate of the green roofs with different vegetation types was *P. prostrates* > *S. lineare* ≈ *P. grandiflora* >* *C. repens* in the wet season, and *P. prostrates* > *S. lineare* > *C. repens* > *P. grandiflora* in the dry season.

The seasonal differences in the recovery rate were largely due to seasonal variations in the weather conditions and vegetation coverage. On the one hand, the solar radiation and temperatures in the dry season were significantly lower than those in the wet season, resulting in a lower ET. On the other hand, the vegetation coverage of the green roofs in the dry season was also lower than that in the wet season (Figure 6), and the average coverage of the green roofs planted with *C. repens*, *P. grandiflora*, *P. prostrates*, and *S. lineare* decreased in the dry season by 14%, 32%, 9%, and 25%, respectively. Both the changes in the weather conditions and vegetation coverage in the dry season led to decreases in the plant transpiration and photosynthetic intensity, and thus, a lower recovery rate.

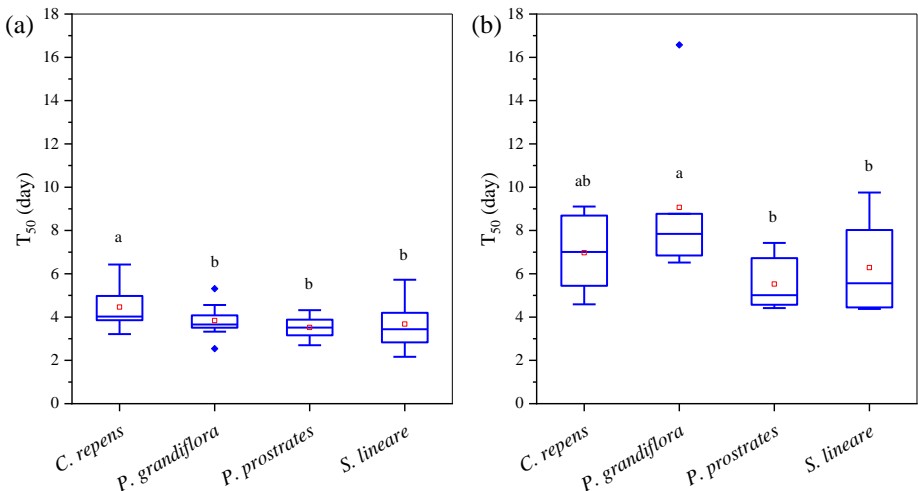

**Figure 5.** Recovery rate ($T_{50}$) for all plant treatments in (**a**) wet season and (**b**) dry season. The letters at the tops of the boxes showed the significances at the 0.05 level (two-way ANOVA without interactions).

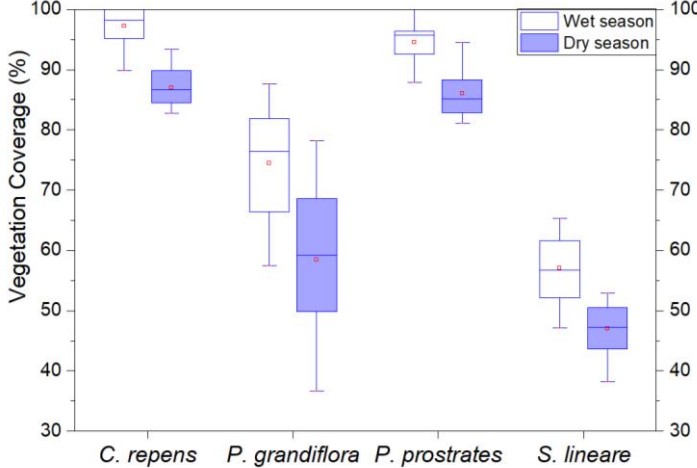

**Figure 6.** Seasonal differences in plant coverage of different types of vegetation.

As for different species treatments, the order of the recovery rates in the different seasons can also be explained by that of the maximum photosynthetic capacity in the corresponding seasons (Figure 4b,c). The seasonal variations in the different plant treatment recovery rates were affected by the changes in the plant coverage. Compared to the wet season, the fitting $T_{50}$ values of the green roofs with *C. repens*, *P. grandiflora*, *P. prostrates*, and *S. lineare* increased by 52%, 139%, 57%, and 75%, respectively, in the dry season. This is nearly consistent with the seasonal variations in the plant coverage under different plant treatments (Figure 6). Compared to the other green roofs, the green roofs with *P. grandiflora* had more obvious seasonal differences in the recovery rates, while the green roofs with *C. repens* had fewer seasonal differences due to their high vegetation coverage throughout the whole monitoring period.

### 3.3. Effect of the Recovery Rate on the Retention Performance of Green Roofs

There is a common notion that a faster recovery rate usually provides a higher capacity for green roofs to retain rainwater from the next rainfall [10,11,16,18]. As mentioned before, the recovery process of green roof rainwater retention is significantly different in different seasons, and the recovery process is faster in the wet season. Figure 7 shows the retention performance in the wet season and dry season between different species treatments, indicating that the wet season, with a faster recovery rate, had a higher retention

volume but a lower retention rate than the dry season. A total of 327–377 mm of rainwater was detained in the wet season, with the rainwater retention percentage ranging from 37 to 43%, while 111–137 mm was retained in the dry season, with the rainwater retention percentage ranging from 45 to 56%. Although the recovery rates of the green roofs were relatively fast in the wet season, the short ADP (52.2 h) made the retention performance of the green roofs unable to recover substantially, resulting in a low rainwater retention rate. The relatively long ADP in the dry season (114.8 h) made it possible to fully recover the retention performance, offsetting the low recovery rate. However, due to the greater total rainfall in the wet season, the retention volume of the rainwater in the wet season was higher than that in the dry season (Figure 7).

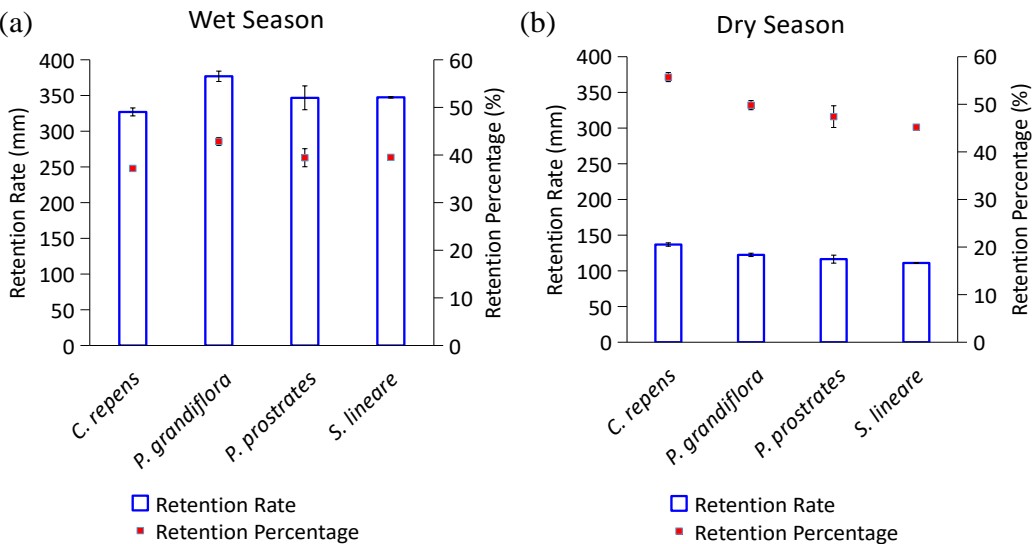

**Figure 7.** The total retention performance of all four plant treatments in the (**a**) wet season and (**b**) dry season. Bars are the standard deviation of the mean.

Figure 7 shows the differences in the rainwater retention capacity among different plant treatments in the two seasons. In the wet season (Figure 7a), the modules planted with *P. grandiflora* had the best rainwater retention performance. The order of the average rainwater retention of the green roofs with different vegetation types was: *P. grandiflora* > * *P. prostrates* > *S. lineare* > *C. repens*, which is consistent with that of the $T_{50}$, except for *P. grandiflora*. This suggests that the $T_{50}$ had a significant effect on the retention performance in the wet season. In addition, the good retention performance of *P. grandiflora* was due to its strong root system (Figure 8), further resulting in a larger actual retention capacity (Figure 9a). Therefore, even though the recovery rates of the green roofs with *P. grandiflora* were slower than those of the other experimental groups, *P. grandiflora* proved to be a good choice for the vegetation layer of the green roofs due to its excellent retention performance.

In the dry season (Figure 7b), the order of the average values of the rainwater retention of green roofs with different vegetation types was: *C. repens* > *P. prostrates* > *P. grandiflora* > *S. lineare.* This order is exactly the same as that of the actual retention capacity in the dry season shown in Figure 9b, instead of the order of the recovery rates. It is worth noting that compared to the wet season, the actual retention capacity of *P. grandiflora* had a significant decline, which may have been due to root atrophy in the dry season. Therefore, the rainwater retention performance of the green roofs was possibly affected by the actual retention capacity in the dry season. Moreover, the order of the actual retention capacity rates of *C. repens*, *P. grandiflora*, and *S. lineare* were similar to those of the plant coverage of the green roofs with different vegetation types in the dry season (Figure 6). This can be explained by the higher plant coverage helping to maintain a larger actual retention capacity by retaining water in the substrate to avoid the water repellency of the

substrate caused by extremely dry conditions [40,41]. However, the *P. prostrates* was an exception. Although the *P. prostrates* had relatively high plant coverage due to its high drought tolerance, it was able to make the soil dry rapidly due to its relatively fast recovery rate, which led to a relatively low retention capacity in the dry season [42].

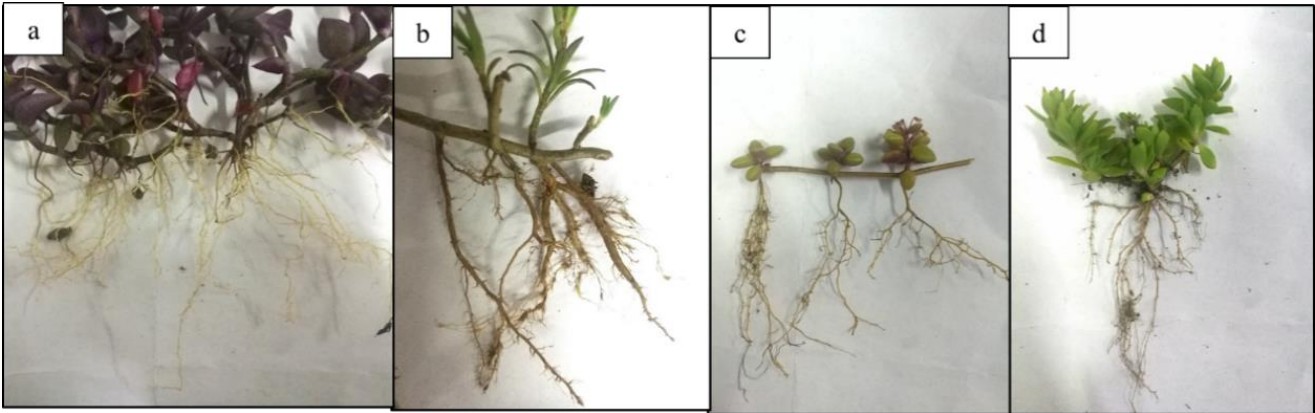

**Figure 8.** Root systems of different plants: (**a**) *C. repens*, (**b**) *P. grandiflora*, (**c**) *P. prostrates*, (**d**) *S. lineare*.

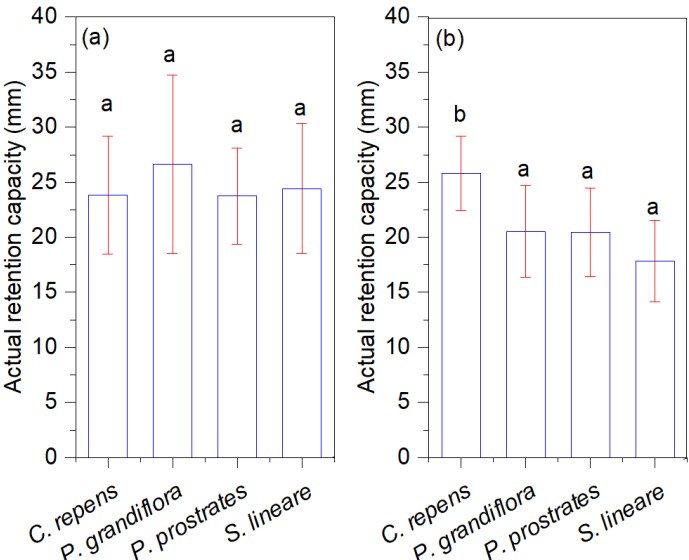

**Figure 9.** Actual retention capacity in (**a**) wet season and (**b**) dry season. The letters at the tops of the columns showed the significances at the 0.05 level (paired *t*-test).

### 3.4. Implication

This study imposed an exponential decline curve to describe the recovery process of green roofs and used $T_{50}$ to represent the recovery rate, providing further insight into the recovery of the rainwater retention capacity during a long period of alternating wet and dry environments. The method helped to accurately evaluate the variation in the recovery among vegetation species and seasons, providing a reference for green roofs in other areas with a typical subtropical monsoon climate similar to Shenzhen.

In the wet season, the faster the recovery was, the better the retention performance of the green roofs was. For the study area region, the $T_{50}$ was used to evaluate the rainwater retention of the green roofs in the wet season. Generally speaking, the green roofs with plants with a high photosynthetic capacity had a faster recovery rate than those with plants with a low photosynthetic capacity. In addition, the plants with strong root systems also enhanced the retention capacity of the green roofs. In order to improve the rainwater retention performance of green roofs in wet seasons, plants with a high maximum

photosynthetic capacity or strong root system should be selected. If only the wet season scenario is considered, *P. prostrates* and *P. grandiflora* should be selected as the preferred green roof plants.

In the dry season, the retention performance of the green roofs was not significantly affected by the recovery rate, and thus, it is unable to be represented by the $T_{50}$. It is important to maintain the life of plants and ensure a certain degree of leaf coverage in green roofs in the dry season, rather than retaining rainwater. Thus, the plants with strong drought resistance should be selected as the green roof plants. In this study, the green roofs with *C. repens* and *P. prostrates* had relatively large plant coverage in the dry season. If only the dry season scenario is considered, these two types of plant should be selected as the preferred green roof plants.

In summary, the optimal green roof plants have the following physiological or ecological characteristics in the study area: in the wet season they should have a high maximum photosynthetic capacity, which leads to a faster recovery rate of the rainwater retention performance, or they should have a strong root system, which leads to a high actual retention capacity. Meanwhile, in the dry season, they should have drought resistance and the ability to maintain relatively large leaf coverage. Therefore, *P. prostrates* is the first choice of green roof plants based on this study.

## 4. Conclusions

In this study, the observations of the rainwater retention performance and its recovery process were conducted for green roofs with four vegetation types (*Callisia repens* Linnaeus, *Portulaca grandiflora* Hook, *Plectranthus prostratus* Gürke, and *Sedum lineare* Thunb) at a site with a typical subtropical monsoon climate. An exponential decline curve was used to describe the recovery process, and the recovery rate was represented by a new indicator, the half-life of the water content in the green roofs after a rain event ($T_{50}$). The observed data were used to analyze the recovery rate of the green roofs with four vegetation types under different seasons and evaluate their effect on rainwater retention performance. The results are summarized as follows:

The decline in the water content after rainfall in the green roofs can be well described by an exponential decline curve ($R^2 > 0.7$), and the average $T_{50}$ values the in green roofs with *Callisia repens* Linnaeus, *Portulaca grandiflora* Hook., *Plectranthus prostratus* Gürke, and *Sedum lineare* Thunb. were 5.5, 5.9, 4.3 and 4.7 days, respectively. The higher the maximum photosynthetic capacity of the green roof plant was, the shorter the $T_{50}$ was.

(1)	The $T_{50}$ in the wet season was significantly shorter than that in the dry season ($p < 0.01$). The $T_{50}$ of the green roofs with *P. grandiflora* had the largest seasonal difference, while the $T_{50}$ of the green roofs with *P. prostrates* had the smallest seasonal difference. The seasonal differences can be explained by the seasonal variations in the weather conditions and eco-physiological activity, such as vegetation species, coverage, and transpiration.

(2)	The rainwater retention rates of the green roofs for rainfall events in the wet season were significantly lower than those in the dry season ($p < 0.05$) due to relatively short antecedent dry periods (ADPs) in the wet season. In the wet season, the retention performance of the green roofs was affected by the $T_{50}$, while in the dry season, the retention performance of the green roofs was mainly determined by the actual retention capacity rather than the $T_{50}$. In addition, a strong plant root system enhanced the retention performance of the green roofs.

(3)	The plants with the following physiological or ecological characteristics are recommended as green roof plants: in wet seasons, they should have a high maximum photosynthetic capacity or a strong root system, while in the dry season, they should have drought resistance and the ability to maintain relatively large vegetation coverage. Therefore, *P. prostrates* was found to be the best choice of green roof plants in this study.

(4)   As an efficient and holistic assessment method, the $T_{50}$ can be used to evaluate the effects of different climates and plants on the recovery process of green roofs based on monitoring data. The results also explain the seasonal differences in weather conditions and plant physiology and provide a reference for selecting suitable green roof plants under a typical subtropical climate. The conclusions of this study are based on the monitoring results at a study area with a typical subtropical monsoon climate, which might be quite different from those in other climate zones. Further studies are needed to investigate the rainwater retention performance of green roofs and their recovery rates in other areas with different climate types and the eco-physiological characteristics of green roof plants. In addition, the applicability of the $T_{50}$ in other scenarios needs further confirmation.

**Author Contributions:** Conceptualization, Y.H., H.Q. and S.-L.Y.; methodology, Y.H.; validation, Y.H. and H.Q.; formal analysis, Y.H.; investigation, Y.H.; resources, Y.H.; data curation, Y.H.; writing—original draft preparation, Y.H.; writing—review and editing, Y.H., H.Q. and Y.O.; visualization, Y.H.; supervision, H.Q.; project administration, Y.H. and H.Q.; funding acquisition, H.Q. All authors have read and agreed to the published version of the manuscript.

**Funding:** This research was supported by the Science and Technology Planning Project of Shenzhen Municipality, China (JCYJ20200109120416654). The APC was funded by Peking University Shenzhen Graduate School.

**Institutional Review Board Statement:** Not applicable.

**Acknowledgments:** The Rainwater Resources Laboratory provided experimental conditions for this paper.

**Conflicts of Interest:** The authors declare no conflict of interest.

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
