# Peer review of "Seasonal Variation in Recovery Process of Rainwater Retention Capacity for Green Roofs"

_water, doi:10.3390/w14182799_

Round 1

Reviewer 1 Report

This paper presents new data on the water usage of green roof plants in a subtropical monsoon climate.  The data collected appears to be robust, and the data analysis appears to be rigorous.  While there are occasional minor errors in English, the manuscript is clearly presented and well illustrated.  The authors show reasonable awareness of relevant literature in the field.

However, there are some significant limitations to the data interpretation, which limit its transferability and generalisation.  As it stands the findings are extremely case specific and not particularly novel or surprising.  These limitations should be addressed prior to publication.

My main reservation about this work is that the authors focus on moisture loss as a function of time, e.g. exponential decay.  However this exponential decay (and also therefore your T50 values) is a function of several things: the local climate (which may fluctuate day-to-day); the substrate depth and moisture holding characteristics; and plant physiology.  If you provide a single curve/relationship, combining all of these factors, then your findings are case-specific and lack transferability/generality. 

A more general relationship should (and can) be derived based on PET, a SMEF and crop factors, as explained below.  The value of such a relationship is that it can be used predictively, to estimate moisture loss (and hence retention capacity) in other climatic contexts.  Your paper would be a far more useful contribution to the literature if you could comment on how well this generic approach fits your own observations.    

The decay is observed as a function of time, but in fact – if the climatic conditions are constant – it is actually a function of available moisture.  As the substrate moisture content reduces, ET falls below Potential ET (PET) in a relationship that is sometimes referred to as a Soil Moisture Extraction Function (SMEF).  Both the Berretta et al. (2014) and the Poe et al. (2015) papers (which you have referenced here) explain this clearly.  The Berretta paper also demonstrates how specific crop factors may be separated out from the effects of local climate and substrate moisture availability.  This would also allow you to explore whether different crop factors are needed to characterise plant behaviours in different seasons, or growth stages.  Again, crop factors have predictive value; this is more useful to scientists and engineers than comments about relative performance – e.g. plant X uses more moisture than plant Y.

There are many internationally-accepted equations that can be used to estimate PET, the most widely-used of which is Penman-Monteith.  You should use your climate data to estimate ET using one or more of the standard formulae.  There is no value in producing new regression relationships that show ET to be a function of temperature; this is already well known and accepted.

Minor comments:

Is the indenting on your highlights correct?  It looks odd.

Lines 64-67.  I am not sure what point you are making here.  The Berretta paper certainly explains a method based on PET, SMEF and crop factors that does allow long term estimates of actual ET, substrate moisture content and retention to be made.  This followed on from the referenced Stovin et al. (2013) paper, which validated the long-term modelling approach against measured rainfall and runoff from a green roof test bed.  The same comment applies to lines 106-109.

Line 136.  ‘vegetable’ should be ‘vegetation’.

Line 144.  A substrate that is 50% peat seems very unusual compared with European or US green roof substrates, where organic content is kept to a minimum to control growth.  Please justify your choice of growing medium.   

Line 189.  Note that while the Poe data did show the exponential decay, that paper argues that actual ET should not be modelled as a function of time, but as a function of PET and soil moisture.

Show cumulative totals of rainfall and runoff over the monitoring period.  If you have continuous data from ~16 months monitoring, readers will be interested to see this, and it will add value to the paper.

Figure 3 needs a key – wet period versus dry period.

Table 1 should be omitted; instead use a standard PET formula.

Section 3.2. needs to be reworked to separate out ‘seasonal’ effects due to climate (PET) from potential crop factor effects.

Reviewer 2 Report

- Please refer to the topic “Instructions for Authors” (https://www.mdpi.com/journal/water/instructions) and adjust references and citations. In line 42, the format of citations should not be [1,2,3] referring to these authors.

- In line 46, add this paper:

Santana, T. C., Guiselini, C., Cavalcanti, S. D. L., da Silva, M. V., Vigoderis, R. B., Júnior, J. A. S., ... & Jardim, A. M. R. F. (2022). Quality of rainwater drained by a green roof in the metropolitan region of Recife, Brazil. Journal of Water Process Engineering, 49, 102953. https://doi.org/10.1016/j.jwpe.2022.102953

- In line 58, add this paper:

Jardim, A. M. R. F., Araújo Júnior, G. N., Silva, M. V., Santos, A., Silva, J. L. B., Pandorfi, H., ... & Silva, T. G. F. (2022). Using Remote Sensing to Quantify the Joint Effects of Climate and Land Use/Land Cover Changes on the Caatinga Biome of Northeast Brazilian. Remote Sensing, 14(8), 1911. https://doi.org/10.3390/rs14081911

- In lines 64-66 add more studies to strengthen the information and affirmative.

- In line 86, add this paper:

Jardim, A. M. R. F., Santos, H. R. B., Alves, H. K. M. N., Ferreira-Silva, S. L., de Souza, L. S. B., Júnior, G. N. A., ... & da Silva, T. G. F. (2021). Genotypic differences relative photochemical activity, inorganic and organic solutes and yield performance in clones of the forage cactus under semi-arid environment. Plant Physiology and Biochemistry, 162, 421-430. https://doi.org/10.1016/j.plaphy.2021.03.011

- In line 137, I believe the species name would be “Portulaca grandiflora Hook.”

- In line 138, I believe the species name would be “Sedum lineare Thunb.”

- In line 153, add an “and” to the ... roof platform; and (b) ...

- In line 193, add an “and” and "the" to ... rain; and k is the recovery coefficient …

- In lines 242-243 please join the text, leaving only a single paragraph.

- Were the models presented in Figure 3 significant?

- In Table 1, please write the species names without capital letters.

- In Figure 4, why not an average test? A test of averages above the bars would be very good. Also, are the bars shown standard error of the mean or standard deviation? Please identify this in the caption.

- In lines 284-286, avoid paragraphs like this. Join the information with the next paragraph.

- In Figure 6, it would be good to standardize the writing of the species. Please write the species names in italics.

- Merge the text of line 330 with line 331.

- In Figure 7, are the bars standard error of the mean, or standard deviation? Please identify this in the caption.

- In Figure 9, was there a significant difference between the species? It would be good to make it clear in the text and, if possible, present some test of average above the bars.

- In lines 411-412 the species are not in italics, please write the name of the species in italics.

Round 2

Reviewer 1 Report

N/A